# Synthesis of Hexagonal Nanophases in the La₂O₃–MO₃ (M = Mo, W) Systems



Egor Baldin [1], Nikolay Lyskov [2,3], Galina Vorobieva [1], Igor Kolbanev [1], Olga Karyagina [4], Dmitry Stolbov [5], Valentina Voronkova [6] and Anna Shlyakhtina [1,*]

[1] N.N. Semenov Federal Research Center for Chemical Physics, Russian Academy of Sciences, Moscow 119991, Russia; baldin.ed16@physics.msu.ru (E.B.); vorob@chph.ras.ru (G.V.); annash@chph.ras.ru (I.K.)

[2] Federal Research Center of Problems of Chemical Physics and Medical Chemistry RAS, Chernogolovka, Moscow 142432, Russia; lyskov@icp.ac.ru

[3] Faculty of Physics, National Research University "Higher School of Economics", Moscow 105066, Russia

[4] Emanuel Institute of Biochemical Physics RAS, Russian Academy of Sciences, Moscow 119334, Russia; olgakar07@mail.ru

[5] Department of Chemistry, Lomonosov Moscow State University, Moscow 119991, Russia; stolbovdn@gmail.com

[6] Department of Physics, Lomonosov Moscow State University, Moscow 119991, Russia; voronk@polly.phys.msu.ru

* Correspondence: annashl@inbox.ru; Tel.: +7-495-939-79-50

**Abstract:** We report a study of nanophases in the La₂O₃–MO₃ (M = Mo, W) systems, which are known to contain a variety of good oxygen-ion and proton conductors. Mechanically activated La₂O₃ + MO₃ (M = Mo, W) mixtures and the final ceramics have been characterized by differential scanning calorimetry (DSC) and X-ray diffraction (XRD) with Rietveld refinement. The microstructure of the materials has been examined by scanning electron microscopy (SEM), and their conductivity in dry and wet air has been determined using impedance spectroscopy. In both systems, the formation of hexagonal La₁₅M₈.₅O₄₈ (phase II, 5H polytype) (M = Mo, W) nanophases is observed for the composition 1:1, with exothermic peaks in the DSC curve in the range ~480–520 °C for La₁₅Mo₈.₅O₄₈ and ~685–760 °C for La₁₅W₈.₅O₄₈, respectively. The crystallite size of the nanocrystalline tungstates is ~40 nm, and that of the nanocrystalline molybdates is ~50 nm. At higher temperatures (~630–690 and ~1000 °C), we observe irreversible reconstructive phase transitions of hexagonal La₁₅Mo₈.₅O₄₈ to tetragonal γ-La₂MoO₆ and of hexagonal La₁₅W₈.₅O₄₈ to orthorhombic β-La₂WO₆. We compare the temperature dependences of conductivity for nanoparticulate and microcrystalline hexagonal phases and high-temperature phases differing in density. Above 600 °C, oxygen ion conduction prevails in the coarse-grained La₁₈W₁₀O₅₇ (phase I, 6H polytype) ceramic. Low-density La₁₅W₈.₅O₄₈ and La₁₅Mo₈.₅O₄₈ (phase II, 5H polytype) nanoceramics exhibit predominantly electron conduction with an activation energy of 1.36 and 1.35 eV, respectively, in dry air.

**Keywords:** mechanochemical synthesis; nanomaterials; lanthanum tungstate; lanthanum molybdate; polytypism; oxygen ion conductivity; proton conductivity





## 1. Introduction

At present, the Ln₂MO₆ (Ln = La-Lu, M = W, Mo) tungstates and molybdates have the greatest potential as host materials for inorganic phosphors, capable of being used as key components of white emitting diodes [1–8]. The synthesis of red-emitting phosphors having near- and middle-UV absorption lines holds significant importance [4,5,9]. This is because when exposed to UV radiation, most materials lose performance and degrade [9].

Urbanovich et al. [10] reported the magnetic properties of the RE₂WO₆ (RE = Nd, Sm, Eu, Gd, Dy, and Ho) tungstates. They found some of them, specifically those with

RE = Nd, Gd, Dy, and Ho, exhibit paramagnetic behavior. The molybdates and tungstates of this family include mixed and oxygen ion conductors [11–13].

In recent years, the $La_2O_3$–$WO_3$ system has attracted increased attention because it was shown to contain fluorite-like $La_{6-x}WO_{12-\delta}$ ($x$ = 0.3–0.7) proton conductors [14–16] with sufficiently high conductivity to be used as proton-conducting SOFC electrolytes and proton-conducting membranes for hydrogen production [17–19]. Along with the $La_{6-x}WO_{12-\delta}$ ($x$ = 0.3–0.7) proton conductors, the $La_2WO_6$ compound [20–22] and other compounds of this system—$La_{18}W_{10}O_{57}$ [23], $La_6W_2O_{15}$ [24], and $La_{10}W_2O_{21}$ [25]—attracted researchers' attention at the same time. A number of studies were concerned with the structure and ionic conductivity of these compounds [24–30]. $\gamma$-$La_6W_2O_{15}$ [24] and $\beta$-$La_2WO_6$ [27,28] were shown to have proton conductivity, but it was two orders of magnitude lower than that of $La_{6-x}WO_{12-\delta}$ ($x$ = 0.3–0.7).

At the same time, the lanthanum tungstate $La_2WO_6$ differs in structural properties from the other RE tungstates having this stoichiometry in that, in the case of a small $La_2O_3$ deficiency, there is a rather narrow range of hexagonal solid solutions ($9La_2O_3$:$10WO_3$) near $La_2WO_6$ ($La_2O_3$:$WO_3$). It is worth noting that there has been discussion for many years as to what compounds exist in the $La_2O_3$–$WO_3$ system near the composition $La_2WO_6$ [23,31–37]. The reason for this is that most compounds in this system have a very narrow or no homogeneity range. Another factor that impedes understanding of this system is that preparing phase-pure ceramics in the tungstate systems requires multiple prolonged anneals and intermediate grindings. Thus, the formation of the RE tungstates in conventional solid-state synthesis is kinetically hindered [38]. It is of interest to note that high-temperature solution growth with a stoichiometric nutrient ($La_2WO_6$ ($La_2O_3$:$WO_3$ = 1:1)) [23,35] yielded hexagonal $La_{18}W_{10}O_{57}$ ($9La_2O_3$:$10WO_3$) single crystals with reduced lanthanum oxide content.

The phenomenon of polytypism is a characteristic feature of these single crystals, according to [23,39]. The $La_{18}W_{10}O_{57}$ single crystals contain two polytypes, 6H (phase I) and 5H (phase II), coexisting in one crystal. The disorder is more pronounced in phase II, which in turn contains slightly more W and O per atom of La. The refined chemical compositions are $La_{18}W_{10}O_{57}$ for phase I (6H polytype ($P\bar{6}c2$, no. 190)) and $La_{15}W_{8.5}O_{48}$ for phase II (5H polytype (P321 no. 150)) [39]. Like single crystals with the same structure, ceramics based on the hexagonal phase $La_{18}W_{10}O_{57}$ (polytype 6H (phase I)) had only oxygen ion conductivity [22,26,28].

The reversible phase transition of $La_2WO_6$ at 1400 °C [35,36] was confirmed in [20–22]. The structures of the low-temperature phase $\beta$-$La_2WO_6$ (orthorhombic symmetry, sp. gr. $P2_12_12_1$, no. 19) and the high-temperature phase $\alpha$-$La_2WO_6$ (orthorhombic symmetry, sp. gr. $Pm2_1n$, no. 31) were completely solved by neutron diffraction [20,21]. The $\alpha$-phase could only be obtained for conductivity measurements by quenching ceramics at 1600 °C, but subsequent heating caused it to transform into the low-temperature phase [21].

The molybdate system $La_2O_3$–$MoO_3$ is distinctive in that there are many fluorite-like phases in its $La_2O_3$-rich part and many scheelite phases in its $MoO_3$-rich part. Note that the structure and properties of some of them are still unexplored [40]. The available structural data for $La_2MoO_6$ are contradictory in many respects [41–44]. Based on the latest crystallographic data reported by Xue et al. [45], who studied the polymorphism of this molybdate by neutron diffraction, it can be concluded that, like $La_2WO_6$, $La_2MoO_6$ exists in two polymorphs: low-temperature phase $\gamma$-$La_2MoO_6$ (tetragonal structure) and high-temperature phase $\alpha$-$La_2MoO_6$ (monoclinic scheelite structure). Note that lanthanum-rich fluorite-like $La_{6-x}MoO_{12-\delta}$ solid solutions were reported to have considerable proton conductivity [46–49]. To the best of our knowledge, there is no evidence for the existence of a molybdate analog of hexagonal $La_{18}W_{10}O_{57}$ (sp. gr. $P\bar{6}c2$, no. 190) or related solid solutions in the $La_2O_3$–$MoO_3$ system.

The comparison of Raman spectroscopy data for ceramics with the nominal composition $La_{18}W_{10}O_{57}$ with Raman spectra of a single crystal also having a hexagonal structure and orthorhombic $\beta$-$La_2WO_6$ ceramic (low-temperature phase) [28], reveals interesting

findings. It demonstrates that the spectrum of the hexagonal crystal is more similar to the spectrum of the β-$La_2WO_6$ ceramic than to the Raman spectrum of the ceramics with the nominal composition $La_{18}W_{10}O_{57}$. This leads us to assume that the β-$La_2WO_6$ ceramic may contain nanodomains of the hexagonal phase $La_{18}W_{10}O_{57}$. It is important to note that, in a previous study [28], ceramics were prepared using mechanical pre-activation of a $La_2O_3$ + $WO_3$ mixture.

Previously, mechanochemical synthesis of the $R_{10}Mo_2O_{21}$ (R = La, Y, Er) molybdates was shown to occur at room temperature [46]. In this process, the crucial factor was structural matching between the starting RE oxide and the synthesized RE molybdate. Thus, given that trigonal $La_2O_3$ ($P3m1$, no. 164) and the hexagonal phase $La_{18}W_{10}O_{57}$ ($9La_2O_3$:$10WO_3$) ($P\overline{6}c2$, no. 190) are structurally similar, mechanical activation of (1) $La_2O_3$ + $WO_3$ and (2) $9La_2O_3$ + $10WO_3$ oxide mixtures would be expected to result in the formation of hexagonal phases at low and even room temperatures. The oxide mixtures (3) $La_2O_3$ + $MoO_3$ and (4) $9La_2O_3$ + $10MoO_3$ are also subjected to mechanical activation in order to obtain hexagonal molybdates. In this way, it will be possible to compare the conductivity of nanoceramics with that of coarse-grained ceramics.

## 2. Materials and Methods

The $La_2MO_6$ (M = Mo, W) materials were prepared using the mechanical activation of starting oxides followed by high-temperature heat treatment of green compacts. The following starting chemicals were used in the synthesis of the lanthanum tungstates and molybdates: $La_2O_3$ (99.99% TU 48-4-523-90, Berdsk, Russia), $WO_3$ (99.9% MRTU 6-09-533-66, Kirovgrad, Russia), and $MoO_3$ (99.9%, Changsha, China). After preheating the starting $La_2O_3$ oxides at 1000 °C for 2 h, they were mixed with the $MoO_3$ or $WO_3$ and co-milled in the high-energy Aronov ball mill [50,51] for 1 h. Oxide mixtures with slightly lower lanthanum oxide content, $9La_2O_3$ + $10WO_3$, were prepared similarly. The mechanically activated mixtures of the oxides were uniaxially pressed at 400–680 MPa and sintered in the range of 600–1540 °C for various lengths of time.

Thermal analysis was performed using the NETZSCH STA 449C system (50–1000 °C or 1200 °C, a heating rate of 10 °C/min, $Al_2O_3$ plate) in the air for mechanically activated oxide mixtures $La_2O_3$ + $MO_3$ and $9La_2O_3$ + $10MO_3$ (M = Mo, W). We performed DSC/TG scans from 50 to 1000 or 1200 °C in an oxygen atmosphere in heating–cooling mode.

X-ray diffraction spectra were obtained at room temperature using the following instruments: (1) the Rigaku Smartlab SE X-ray diffractometer and (2) the DRON-3M automated diffractometer. The SmartLab Studio II software was used for Rietveld refinement and crystallite size determination. The recording modes were as follows: (1) Cu Kα radiation, λ = 1.5418 Å, Bragg-reflection geometry, 40 kV, 50 mA; 2θ range was 10° to 130°, scan step 0.005°, scan rate 3°/min) in continuous mode and (2) Cu Kα radiation, λ = 1. 5418 Å, Bragg-reflection geometry, 35 kV, 28 mA; 2θ range from 10° to 75° (scan step of 0.1°, τ = 3 s), respectively. High-temperature annealed $La_{18}W_{10}O_{57}$ powder (1400 °C, 4 h) was used as a reference to determine the instrumental line broadening. Scanning electron microscopy (SEM) (JEOL JSM-6390LA, Tokyo, Japan) was used to investigate the microstructure of the powder and ceramic samples. Platinum ink (ChemPur C3605, Karlsruhe, Germany) was applied to the opposite sides of the ceramic tablets. Further annealing was carried out at 650–800 °C for 30 min. A P-5X potentiostat/galvanostat combined with a frequency response analyzer module (Electrochemical Instruments, Ltd., Saint Petersburg, Russia) was used for dry and wet air electrical conductivity measurements. Details of the conductivity measurements are given in [16,46].

## 3. Results and Discussion

### 3.1. Particle Size and Morphology of Starting Oxide Powders and Mechanically Activated Powders

Figure 1 illustrates the particle size and morphology of the starting oxides: (a) $La_2O_3$, (b) $WO_3$, and (c) $MoO_3$. $La_2O_3$ is characterized by a wide range of particle sizes and the formation of large aggregates in the form of irregular polyhedrons whose dimensions can

exceed 3 μm. The smallest particles are 50–100 nm and nearly spherical. $WO_3$ consists of homogeneous spheres with an average size of ~150 nm. $MoO_3$ particles in the form of thin plates (thickness~500 nm and length~2 μm) are the most crystallized among the three starting oxides.

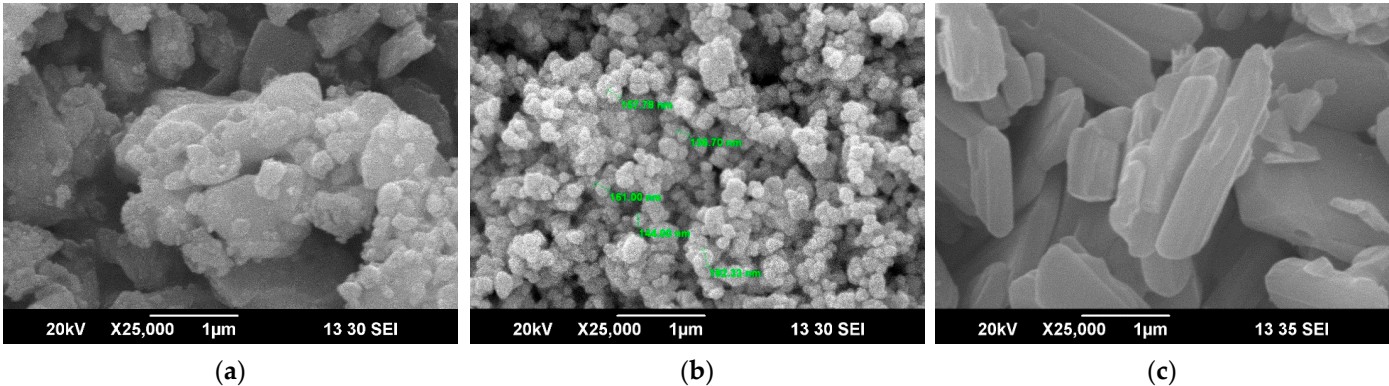

**Figure 1.** Microstructure of the starting oxides: (**a**) $La_2O_3$, (**b**) $WO_3$, and (**c**) $MoO_3$.

Figure 2 shows the morphology and cation distribution in the $La_2O_3 + WO_3$ powder mixture after mechanical activation (m/a). Figure 2a shows the powder particle size of the $La_2O_3 + WO_3$ oxide mixture after ball milling. The powder particles have an average size of ~0.5–2 μm. The shape of the particles is almost spherical. Figure 2c,d illustrates the cation distribution in the mechanically activated powder $La_2O_3 + WO_3$ (magnification 200 μm). One can see a uniform distribution of La and W cations in the final m/a powder.

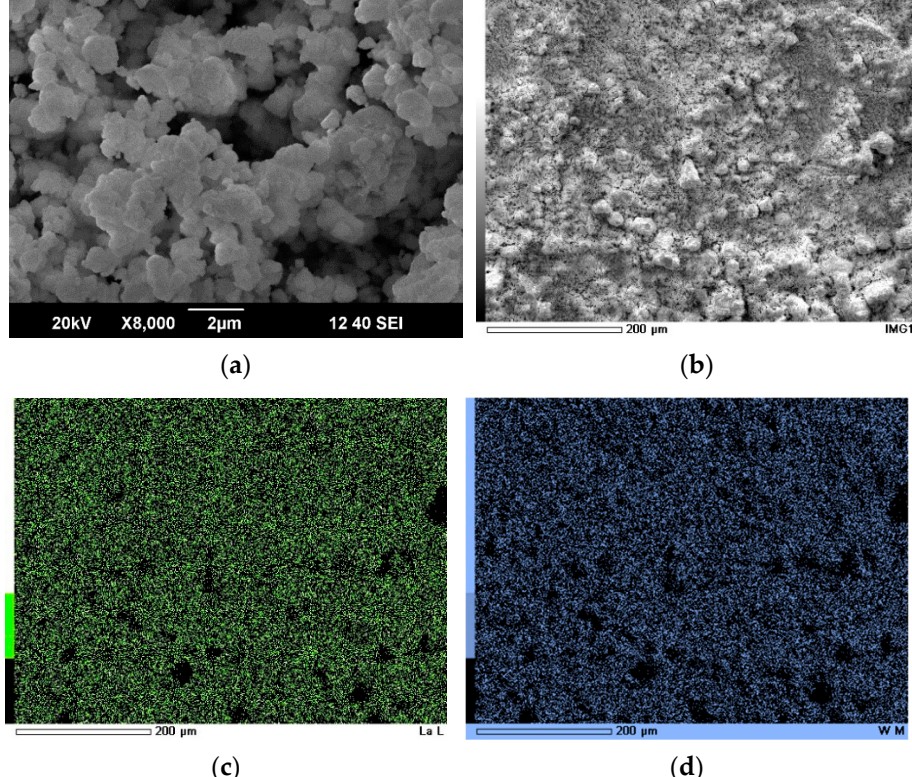

**Figure 2.** (**a**) SEM image (magnification 2 μm) and (**b**) SEM image (magnification 200 μm) (**c,d**) La and W X-ray maps, respectively, of the mechanically activated (m/a) $La_2O_3 + WO_3$ oxide.

The morphology and cation distribution of the $La_2O_3$ + $MoO_3$ powder mixture after mechanical activation are shown in Figure 3. Figure 3a shows that the powder particles have an average size of ~0.5–1.5 µm after milling. The particle shape is also close to spherical. Figure 3b–d illustrates the cation distribution in the mechanically activated powder (magnification 200 µm). A uniform distribution of La and Mo cations can be seen in the final m/a powder.

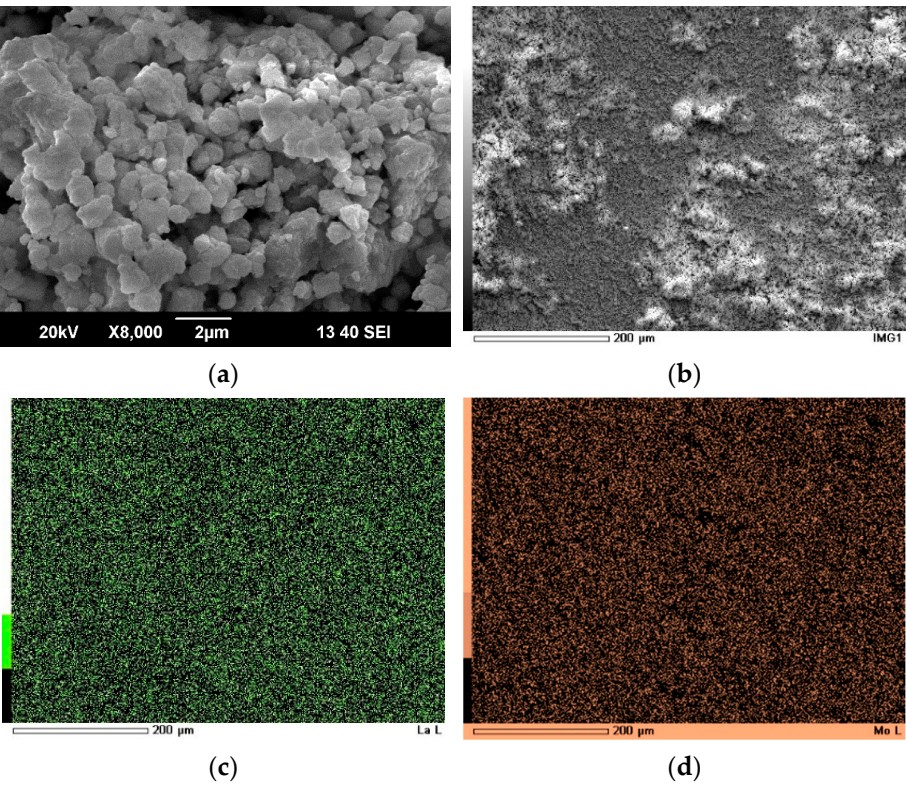

**Figure 3.** (**a**) SEM image (magnification 2 µm) and (**b**) SEM image (magnification 200 µm) (**c**,**d**) La and Mo X-ray maps of the mechanically activated (m/a) $La_2O_3$ + $MoO_3$ oxide mixture, respectively.

### 3.2. Synthesis of the Hexagonal Phase $La_{18}W_{10}O_{57}$ ($9La_2O_3{:}10WO_3$) from a Mechanically Activated $La_2O_3$ + $WO_3$ Mixture

Figure 4a presents the results for a mechanically activated $La_2O_3$ + $WO_3$ mixture during heating in a DSC cell to 1000 °C and subsequent cooling. The DSC curve shows a single exotherm in the 685–760 °C range with a peak temperature of ~730 °C (Figure 4a). To identify its origin, the same powder was heated twice: (1) to 685 °C and (2) to 760 °C. The XRD results for the two samples are presented in Figure 4b, scans 1 and 2, respectively.

The powder was X-ray amorphous after firing in the DSC cell at 685 °C. However, its XRD pattern showed some lines attesting to an onset of crystallization (Figure 4b, scan 1). After firing at 760 °C, we observed broad lines of the hexagonal phase $La_{18}W_{10}O_{57}$ ($9La_2O_3{:}10WO_3$) (Figure 4b, scan 2). Table S1 and Figure S1 present the X-ray structure analysis (Rietveld refinement) results for this sample. These demonstrate that the major phase at 760 °C is well described by the $La_{18}W_{10}O_{57}$ (phase I, 6H polytype, sp. gr. $P\bar{6}c2$, no. 190) structure model. Table S1 presents structure analysis results obtained for this sample using Rietveld refinement in a two-phase ($La_{18}W_{10}O_{57}$ + $La_2O_3$) model, which turned out slightly poorer. Next, a mechanically activated starting oxide mixture was subjected to prolonged (96 h) firing at 600 °C. Figure 4b (scan 3) presents the XRD results for this sample. Under these conditions, we obtained a two-phase mixture of $La_{18}W_{10}O_{57}$ ($9La_2O_3{:}10WO_3$) and $La_2O_3$. The crystallization temperature determined using a DSC cell is known to be overestimated because of the high heating rate of the powder. In contrast, long annealing times lead to the crystallization of compounds at a lower temperature [52].

In our case, this occurs at 600 °C, and we observe the presence of a second crystalline phase: $La_2O_3$. Therefore, we conclude that the rate of room-temperature mechanochemical synthesis in the case of the lanthanum tungstate is lower than in the case of the $La_{10}Mo_2O_{21}$ molybdate [46] and that a considerable fraction of the lanthanum oxide does not react with $WO_3$.

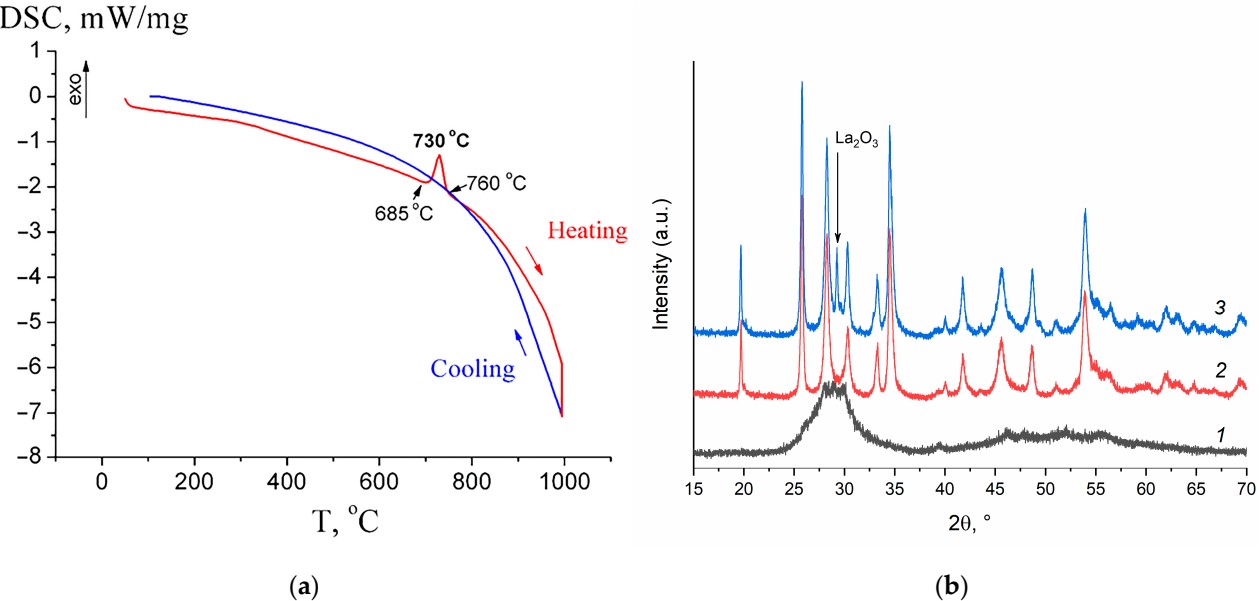

**Figure 4.** (**a**) Data obtained in a DSC cell during heating to 1000 °C, illustrating the exothermic process in a mechanically activated (m/a) $La_2O_3 + WO_3$ mixture. (**b**) XRD patterns of mechanically activated (m/a) $La_2O_3 + WO_3$ mixture (1) after heating in DSC cell up to 685 °C, (2) after heating in DSC cell up to 760 °C, and (3) after thermal annealing at 600 °C for 96 h. $La_2O_3$ (ICDD PDF 1523968): −41.1(6) wt.%.

Considering that the low-temperature annealed phase is nanoscale and probably disordered, we carried out an additional study over a wider range of angles from 10–130°. We refined the structure using a more disordered hexagonal phase $La_{15}W_{8.5}O_{48}$ model (phase II, 5H polytype (*P*321 no. 150)) [39]. The result of Rietveld's refinement is shown in Figure 5. The model $La_{15}W_{8.5}O_{48}$ (phase II, 5H polytype) describes the nanosized hexagonal phase better than the model $La_{18}W_{10}O_{57}$ (phase I, 6H polytype) (Table S2). It is important to note that due to the broadening of the diffraction lines caused by the nanoscale of the samples, the differences in the diffraction profiles of both these polytypes are minimal.

Therefore, the exothermic peak in the DSC curve (Figure 4a) is due to the crystallization of the hexagonal phase $La_{15}W_{8.5}O_{48}$ from an X-ray amorphous state. The crystallization process is irreversible, as can be seen from the thermal analysis data in Figure 4a.

Figure 6a,b shows powder XRD patterns of disk-shaped compacts produced by pressing a mechanically activated starting $La_2O_3 + WO_3$ mixture and firing at various temperatures. It can be seen that the hexagonal phase exists in the range 800–900 °C, whereas at 1000 °C and higher temperatures, up to 1540 °C, the major phase is orthorhombic β-$La_2WO_6$. However, results of a more detailed study indicated that in the range 1000–1540 °C, the ceramic contained trace levels of the hexagonal phase ($9La_2O_3:10WO_3$), ranging from 4 to 9%, depending on firing conditions. At 1620 °C, the ceramic melted.

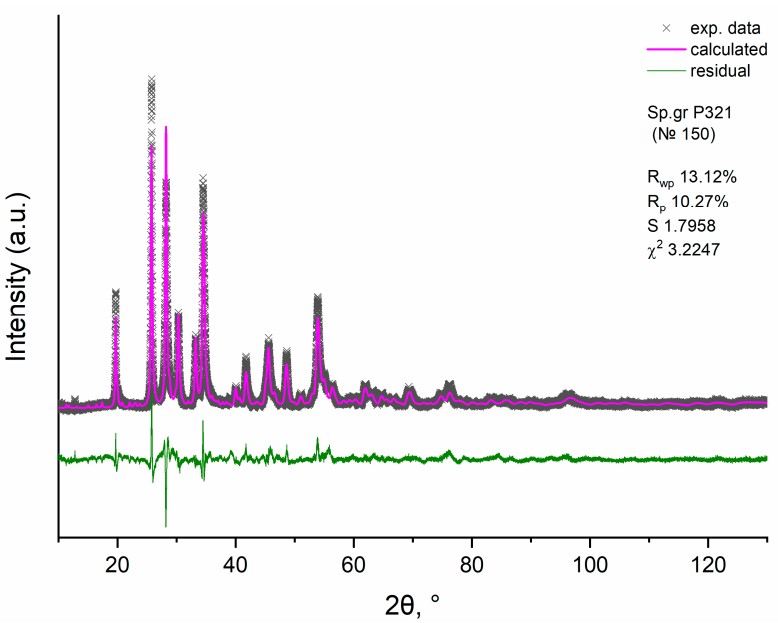

**Figure 5.** Rietveld refinement of the XRD pattern of the m/a $La_2O_3$ + $WO_3$ mixture after annealing at 800 °C for 4 h. Crosses represent the observed pattern; the solid line through the symbols shows the calculated fit. The difference between the observed and calculated data is shown at the bottom.

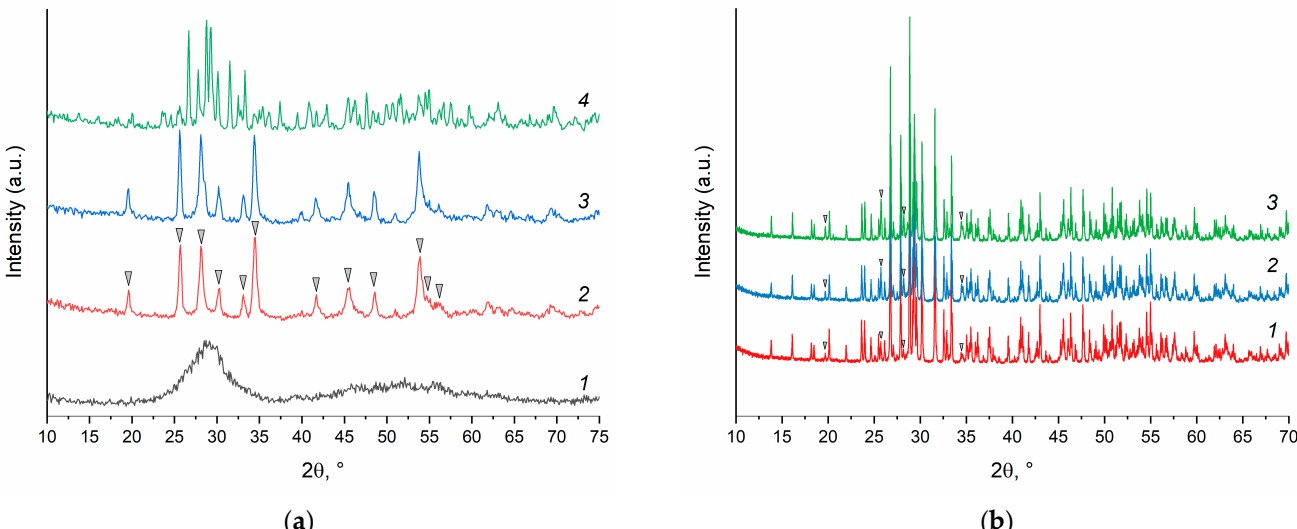

(**a**)            (**b**)

**Figure 6.** (**a**) XRD patterns of the ceramics produced by firing a mechanically activated $La_2O_3$ + $WO_3$ mixture in the range of 25–1000 °C, illustrating the formation of the hexagonal phase in the range of 800–900 °C and the phase transition to the orthorhombic phase β-$La_2WO_6$: (1) $La_2O_3$ + $WO_3$ starting mixture after mechanical activation; (2) 800 °C, 6 h; (3) 900 °C, 6 h; (4) 1000 °C, 6 h. (**b**) XRD patterns of the ceramics produced by firing a mechanically activated $La_2O_3$ + $WO_3$ mixture in the range 1400–1500 °C: (1) 1400 °C, 4 h; (2) 1000 °C, 6 h; 1350 °C, 2 h; 1420 °C, 4 h; (3) 1500 °C, 2 h. The triangles mark reflections from the hexagonal phase (9$La_2O_3$:10$WO_3$).

    No signs of the high-temperature phase α-$La_2WO_6$ were detected, which is unsurprising because the transition at ~1400 °C is reversible [21]. To summarize the above, we used mechanical activation to synthesize lanthanum tungstate with the nominal composition $La_2WO_6$ ($La_2O_3$:$WO_3$) from an oxide mixture. We obtained a hexagonal $La_{15}W_{8.5}O_{48}$ nanophase at low temperatures. In contrast, in the range of 1000–1540 °C at higher temperatures, the orthorhombic phase β-$La_2WO_6$ was found to coexist with trace levels of the hexagonal phase (9$La_2O_3$:10$WO_3$). It should be emphasized that the hexagonal-to-orthorhombic phase transition did not show up in the DSC curve of a $La_2O_3$ + $WO_3$ mixture

(Figure 4a) but was detected during isothermal holding (Figure 6a). It is reasonable to assume that it is a second-order transition.

### 3.3. Synthesis of the Hexagonal Phase $La_{18}W_{10}O_{57}$ from a Mechanically Activated $9La_2O_3$ + $10WO_3$ Mixture

Figure 7a,b presents XRD results for the ceramics prepared by firing a mechanically activated oxide mixture ($9La_2O_3$ + $10WO_3$) in the range of 800–1620 °C. It is seen (Figure 7a, scans 1–5) that the hexagonal phase exists in the range of 800–1400 °C. X-ray structure analysis (with Rietveld refinement) of the ceramic prepared by firing at 1400 °C for 4 h and then ground into powder (Figure 7b) showed that the material was single-phase and consisted of the hexagonal phase $La_{18}W_{10}O_{57}$ (polytype 6H, phase I, sp. gr. $P\bar{6}c2$, no. 190; Table S2). After annealing at low temperatures of 800 °C and 900 °C, the X-ray peaks of the hexagonal phases are significantly broadened (Figure 7a, scans 1 and 2), and we cannot unequivocally identify the polytype to which they belong. Assuming that the 5H polytype is more disordered, it is probably realized in this case. Thus, the hexagonal phase $La_{18}W_{10}O_{57}$ ($9La_2O_3$:$10WO_3$) has a narrow homogeneity range. At the initial nominal composition of $La_2WO_6$ ($La_2O_3$ + $WO_3$), after firing in the range of 1000–1400 °C, this phase was detected by XRD in a two-phase region: $\beta$-$La_2WO_6$ (major phase) and $La_{18}W_{10}O_{57}$ (impurity phase) (Figure 6b). High-temperature firing at 1620 °C led to the formation of an additional phase due to the partial melting of the ceramic (Figure 7a, scan 6).

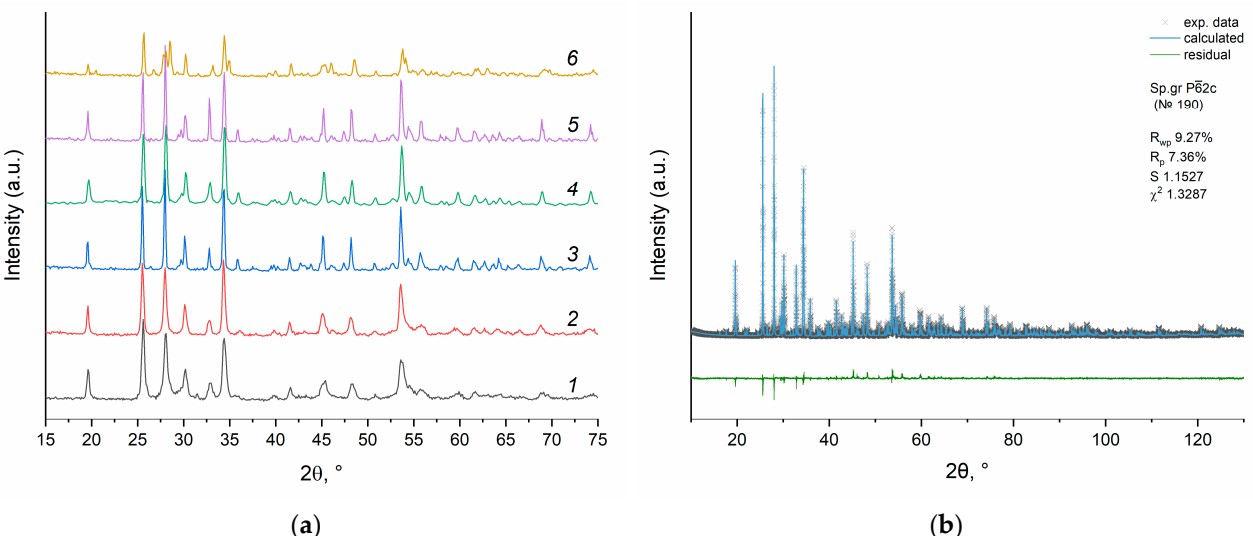

**Figure 7.** (**a**) XRD patterns of the ceramics produced by firing a mechanically activated $9La_2O_3$ + $10WO_3$ mixture in the range of 800–1620 °C: (1) 800 °C, 6 h; (2) 900 °C, 6 h; (3) 1200 °C, 4 h; (4) 1300 °C, 4 h; (5) 1400 °C, 4 h; (6) 1620 °C, 1 h. (**b**) Rietveld refinement of the XRD pattern of the $La_{18}W_{10}O_{57}$ (polytype 6H, phase I) sample synthesized at 1400 °C for 4 h. Crosses represent the observed pattern, and the solid line through the symbols shows the calculated fit. The difference between the observed and calculated data is shown at the bottom.

The $La_{15}W_{8.5}O_{48}$ (phase II polytype 5H; Table S2) nanoceramic was prepared by firing a $La_2O_3$ + $WO_3$ mixture at 800 °C for 6 h. The mixture was then ground into powder and studied in a DSC cell during heating and cooling in a broader temperature range, up to 1200 °C (Figure S2a). Under the same conditions, we studied the coarse-grained $La_{18}W_{10}O_{57}$ (phase I polytype 6H; Table S2) ceramic prepared by firing a $9La_2O_3$ + $10WO_3$ mixture at 1400 °C for 4 h and then ground into powder (Figure S2b). In both cases, an exotherm in the range of ~800–1040 °C was detected. It is most likely that this effect is inherent to the hexagonal phase with two polytypes itself [39]. XRD characterization showed that after heating to 1200 °C and cooling in a DSC cell, the hexagonal nanophase produced by firing a $La_2O_3$ + $WO_3$ mixture at 800 °C for 6 h consisted of two phases:

β-La$_2$WO$_6$ (82.5%) and hexagonal La$_{18}$W$_{10}$O$_{57}$ (17.5%). XRD characterization of the coarse crystalline hexagonal La$_{18}$W$_{10}$O$_{57}$ phase after heating to 1200 °C and cooling in a DSC cell showed a 100% hexagonal phase.

### 3.4. Synthesis of a Hexagonal Phase in the La$_2$O$_3$–MoO$_3$ System from a Mechanically Activated Oxide Mixture

A mechanically activated La$_2$O$_3$ + MoO$_3$ mixture was also studied during heating to 1000 °C in a DSC cell and cooling. Its DSC curve showed two exotherms: (1) in the range of 480–520 °C, with a peak temperature of ~500 °C, and (2) an appreciably larger peak in the range of 630–690 °C, with a peak temperature of 665 °C (Figure 8a). Both events were irreversible and were not detected during cooling. With allowance for the DSC data, we performed a series of 4 h isothermal heat treatments of disk-shaped compacts at temperatures below and above these peaks: 400, 500, 600, 700, 800, and 900 °C.

Figure 8b presents XRD data for the ceramics obtained at these temperatures and ground into powder. The La$_2$MoO$_6$ sample prepared by firing at 600 °C for 4 h also consisted of a hexagonal phase, detected for the first time in the La$_2$O$_3$–MoO$_3$ system. Rietveld refinement with molybdenum instead of tungsten yielded a structure similar to the hexagonal structure of La$_{15}$W$_{8.5}$O$_{48}$ (Figure 9, Table S2). However, note that the line at 2θ~36.5° is inconsistent with this structure.

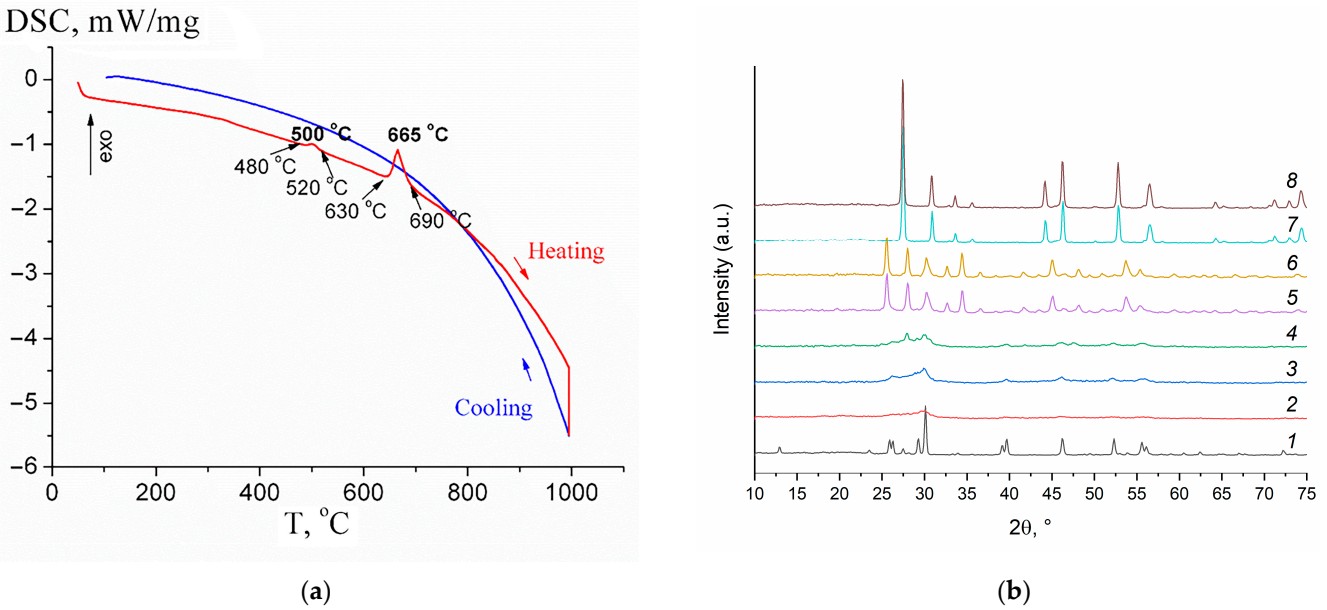

**Figure 8.** (**a**) Data obtained in a DSC cell during heating to 1000 °C illustrate the exothermic process in a mechanically activated La$_2$O$_3$ + MoO$_3$ mixture. (**b**) XRD patterns of a La$_2$O$_3$ + MoO$_3$ mixture (1) before and (2) after mechanical activation and after firing in the range of 400–900 °C: (3) 400 °C, 4 h; (4) 500 °C, 4 h; (5) 600 °C, 4 h; (6) 700 °C, 4 h; (7) 800 °C, 4 h; (8) 900 °C, 4 h.

Next, a mechanically activated starting oxide mixture was subjected to prolonged (48 h) firing at 600 °C to ascertain whether La$_2$O$_3$ impurities were present at this temperature. After prolonged (100 h) holding, a small peak of γ-La$_2$MoO$_6$ emerged at 600 °C as well (3.7 wt.%), but no traces of La$_2$O$_3$ were detected. Clearly, the mechanochemical synthesis of La molybdate at room temperature was easier than that of La tungstate. Moreover, due to diffusion hindrances, the solid-state synthesis of tungstates is known to take longer than the synthesis of molybdates [38].

It should be emphasized that if the hexagonal lanthanum molybdate La$_{15}$Mo$_{8.5}$O$_{48}$ (phase II, polytype 5H (P321 no. 150)) was detected, the ceramic was annealed at a temperature (500–700 °C) below the melting point of molybdenum oxide, which is 795 °C for pure MoO$_3$. After 800 °C firing, tetragonal γ-La$_2$MoO$_6$ was the major phase. A certain amount of this phase (10.3 wt.%, with the strongest line present) was detected even after 700 °C

firing (Figure 8b, scan 6). We believe that the lower temperature exothermic peak in the DSC curve, in the range of 480–520 °C, is due to the crystallization of the hexagonal phase $La_{15}Mo_{8.5}O_{48}$ (phase II, polytype 5H) from an X-ray amorphous state. Additionally, the higher temperature exothermic peak is due to the transition of the hexagonal nanophase to the tetragonal γ-$La_2MoO_6$ phase. Clearly, this is a first-order reconstructive transition (Figure 8a).

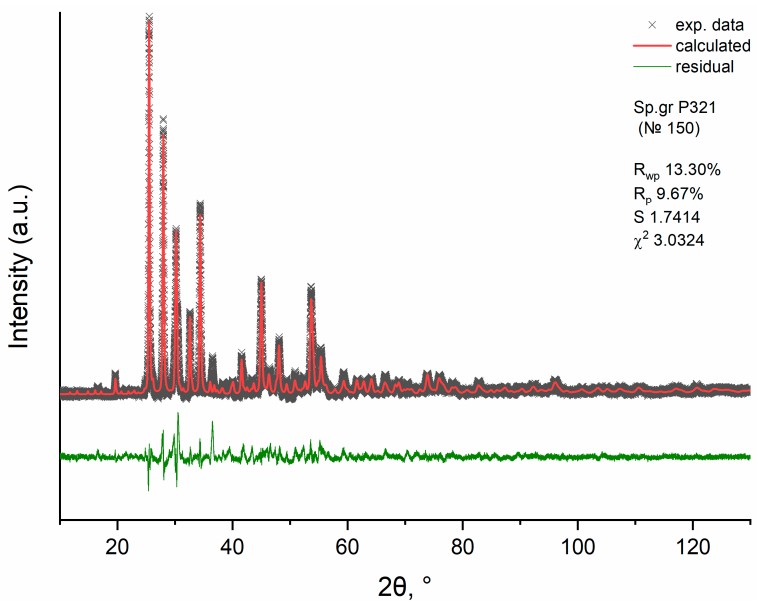

**Figure 9.** Rietveld refinement of the XRD pattern of the m/a $La_2O_3$ + $MoO_3$ mixture after firing at 600 °C for 4 h. Crosses represent the observed pattern. The solid line through the symbols shows the calculated fit. The difference between the observed and calculated data is shown at the bottom.

### 3.5. Evaluation of the Crystallite Size in Hexagonal Lanthanum Tungstate and Lanthanum Molybdate and Microstructure of the Nanoceramics

From the width of the 116 lines of $La_{15}W_{8.5}O_{48}$ (2θ~25.7°), we can formally estimate the crystallite size L of the hexagonal nanophases using the Scherrer formula.

Such an estimate was made for hexagonal $La_{15}W_{8.5}O_{48}$ obtained at 800 and 900 °C (Figure 6a, scans 2 and 3) and hexagonal $La_{15}Mo_{8.5}O_{48}$ synthesized at 600 and 700 °C (Figure 8b, scans 5 and 6). The crystallite size of the lanthanum tungstate was determined to be 37.6 and 42 nm after firing at 800 and 900 °C, respectively. The lanthanum molybdate was 47.6 and 49.8 nm after firing at 600 and 700 °C, respectively. Thus, the crystallite size of the lanthanum molybdate was slightly larger, even though it was synthesized at lower temperatures than the tungstate.

This result confirms that, during room-temperature mechanochemical synthesis, $La_2O_3$ is more reactive with molybdenum oxide than tungsten oxide. The solid-state reaction between the starting oxides occurs more readily in the case of the RE molybdates, which can be synthesized even via room-temperature mechanochemical processing [46]. Accordingly, $La_{15}Mo_{8.5}O_{48}$ subsequently crystallizes more readily than $La_{15}W_{8.5}O_{48}$. The mechanochemical synthesis of $La_{15}W_{8.5}O_{48}$ does not reach completion at room temperature, and the mixture contains unreacted $La_2O_3$. This unreacted $La_2O_3$ eventually crystallizes during prolonged 600 °C firing (Figure 4b, scan 3).

The SEM data in Figure 10a,b demonstrate that the hexagonal $La_{15}Mo_{8.5}O_{48}$ and $La_{15}W_{8.5}O_{48}$ ceramics are nanograined. It is worth noting that, unlike the $La_{15}Mo_{8.5}O_{48}$ ceramic, the $La_{15}W_{8.5}O_{48}$ nanoceramic exhibits nanograin aggregation. This is possibly due to the higher synthesis temperature of lanthanum tungstate. The result of applying the anisotropic model to calculate crystallite size was unsuccessful.

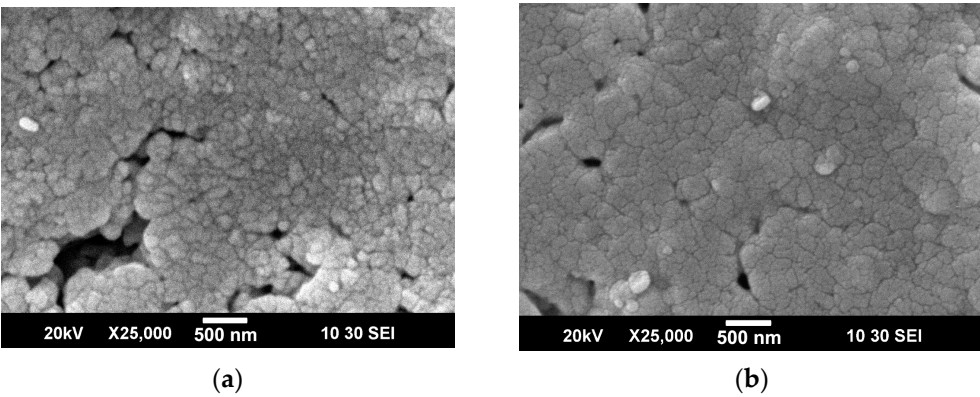

(**a**)

(**b**)

**Figure 10.** Microstructure of (**a**) the $La_{15}Mo_{8.5}O_{48}$ nanoceramic prepared at 700 °C (4 h) and (**b**) the $La_{15}W_{8.5}O_{48}$ nanoceramic prepared at 900 °C (6 h).

*3.6. Ionic Conductivity of the Hexagonal $La_{15}M_{8.5}O_{48}$ (M = Mo, W) Nanophases, Coarse-Grained $La_{18}W_{10}O_{57}$ Ceramic, and High-Temperature Phases (Orthorhombic $\beta$-$La_2WO_6$ and Tetragonal $\gamma$-$La_2MoO_6$)*

In comparing the total conductivity of the low-density hexagonal $La_{15}W_{8.5}O_{48}$ nanoceramic prepared by firing at 800 °C for 6 h (relative density of 60.0%) (Table S3) and that of the previously studied [28] coarse-grained $La_{18}W_{10}O_{57}$ ceramic (Figure S3) prepared by firing at 1400 °C for 4 h (relative density of 92%) (Figure 11, plots 1 and 2), it is worth noting that these materials have different Arrhenius plots (obtained by impedance spectroscopy).

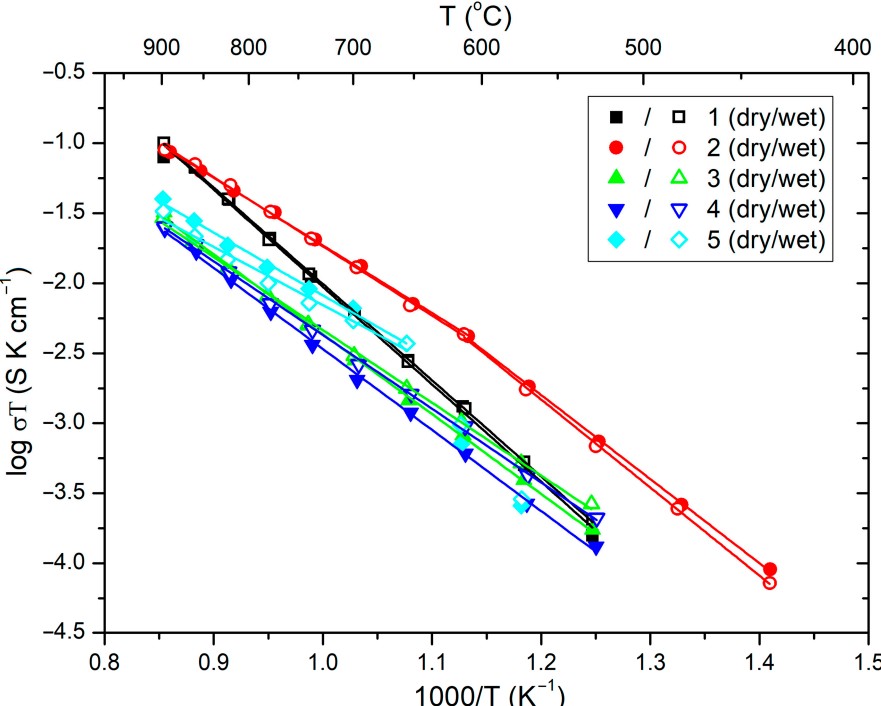

**Figure 11.** Arrhenius plots of total conductivity for LaWO materials in dry (filled data points) and wet (open data points) air: (1) low-density hexagonal $La_{15}W_{8.5}O_{48}$ nanoceramic (relative density of 60.0%) produced by firing a mechanically activated $La_2O_3 + WO_3$ mixture at 800 °C for 6 h, (2) dense coarse-grained $La_{18}W_{10}O_{57}$ ceramic (relative density of 92%) produced by firing a mechanically activated $9La_2O_3 + 10WO_3$ mixture at 1400 °C for 4 h, (3) low-density orthorhombic $\beta$-$La_2WO_6$ ceramic (relative density of 69.6%) produced at 1000 °C (6 h) from a mechanically activated $La_2O_3 + WO_3$ mixture, (4) coarse-grained orthorhombic $\beta$-$La_2WO_6$ ceramic (relative density of 81.4%) produced at 1400 °C (4 h) from a mechanically activated $La_2O_3 + WO_3$ mixture, and (5) $La_2W_{1+x}O_{6+3x}$ ($x\sim0.22$) single crystal [28].

The activation energy for oxygen ion conduction between 400 and 600 °C in dry air is 1.36 and 1.19 eV in the nanoceramics ($La_{15}W_{8.5}O_{48}$) and coarse-grained ceramics ($La_{18}W_{10}O_{57}$), respectively (Table 1). Above 600 °C, the activation energy of the nanoceramics changes insignificantly, whereas that of the coarse-grained ceramics decreases to $E_a = 0.96$ eV (Table 1). It seems likely that, in the loose $La_{15}W_{8.5}O_{48}$ nanoceramic, electronic conduction prevails in the air. This is supported by the lack of electrode dispersion in its impedance spectrum (Figure S4a), which contrasts with the spectrum of the coarse-grained $La_{18}W_{10}O_{57}$ ceramic (Figure S4b).

**Table 1.** Apparent activation energy ($E_a$) of total conductivity temperature dependences of LaWO materials in dry and wet air.

| No. | Composition, $T_{synthesis}$ | Atmosphere | $E_a$ ($\pm 0.01$), eV | |
| --- | --- | --- | --- | --- |
| | | | 400–600 °C | 600–900 °C |
| 1 | $La_{15}W_{8.5}O_{48}$–800 °C | Dry air | 1.36 | |
| | | Wet air | 1.35 | |
| 2 | $La_{18}W_{10}O_{57}$–1400 °C | Dry air | 1.19 | 0.96 |
| | | Wet air | 1.25 | 0.98 |
| 3 | $\beta$-$La_2WO_6$–1000 °C | Dry air | 1.13 | |
| | | Wet air | 1.03 | |
| 4 | $\beta$-$La_2WO_6$–1400 °C | Dry air | 1.16 | |
| | | Wet air | 1.05 | |
| 5 | $La_2W_{1+x}O_{6+3x}$ ($x\sim0.22$) single crystal | Dry air | - | 0.89 |
| | | Wet air | - | 0.83 |

As to the nature of charge transport in the LaWO compounds studied, we found that the size affected the conductivity type. In the case of ceramics consisting of micron-sized grains, under oxidizing conditions, where W is in its highest oxidation state, 6+, ionic (oxygen ion) conductivity prevails. At the same time, some compositions may have proton transport at a certain ambient humidity, depending on their structure type. This is supported by the shape of the impedance response (e.g., in Figure 4b), which is characterized by dispersion at high frequencies, where effects related to bulk ionic transport show up, and low-frequency dispersion (the electrode component of the impedance), typical of gas-phase exchange processes with charge transport across the electrode/electrolyte interface. Nanoceramics have a different impedance response and a different dominant contribution to conductivity, which determines their type. The impedance data for the nanoceramics (Figure S4a,c) lead us to assume that there is an electronic component of conductivity, which shows up as the absence of low-frequency electrode impedance dispersion. We then observe an impedance response characteristic of bulk and grain-boundary processes. Thus, it is reasonable to assume that mixed electronic–ionic conduction prevails in nanoceramics.

As to the possible mechanism of the mixed electronic–ionic conduction in this compound, note that it occurs predominantly under reducing conditions, which reduce W to the oxidation state 5+ ($W^{6+} + e' \rightarrow W^{5+}$). Electronic carriers may result from partial oxygen loss under reducing conditions, according to the scheme:

$$0 \Leftrightarrow V_O^{\bullet\bullet} + 1/2 O_2 + 2e' \tag{1}$$

The resulting electrons can participate in electron transport processes, localizing at *W*. It seems likely that, in nanoceramics, local changes in the charge state of *W* on the grain surface are possible, which can result in mixed electronic–ionic conduction. Obviously, $La_{15}W_{8.5}O_{48}$ nanophase is a mixed electron–ionic conductor, but electronic conduction prevails in the air. Plot 5 in Figure 11 is the Arrhenius plot of conductivity for a previously studied hexagonal $La_2W_{1+x}O_{6+3x}$ ($x\sim0.22$) single crystal [28].

It is seen (Figure 11, plots 2, 5) that above 600 °C, the coarse-grained ceramic and hexagonal single crystal differ little in $E_a$, 0.96 and 0.89 eV, respectively (Table 1). This suggests an oxygen vacancy conduction mechanism. As the grain size decreases to a nanoscale, surface electron conduction prevails, and the activation energy rises considerably (Table 1; Figure 11, plot 1). Note that below 600 °C, the conductivity of the hexagonal $La_2W_{1+x}O_{6+3x}$ ($x\sim0.22$) single crystal drops sharply, which is accompanied by an increase in activation energy to 2.17 eV. The origin of this effect is not yet fully clear.

A comparison of the proton conductivity of the high-temperature, orthorhombic β-$La_2WO_6$ ceramics differing in relative density (69.6 and 81.4%) (Figure 11, plots 3, 4) shows that in wet air, they have essentially the same proton conductivity. This is $\sim1 \times 10^{-6}$ S/cm at 600 °C, independent of the density of the ceramics. This finding supports the conclusion drawn by Fleig [53] in a theoretical study concerned with modeling the impedance of highly porous materials. They found that the porosity of ceramics has little effect on the temperature dependence of their bulk conductivity derived from impedance spectroscopy data. The activation energies for conduction in dry air ($E_a$ = 1.13 и 1.16 eV) and wet air ($E_a$ = 1.03 и 1.05 eV) differ little and are independent of the density of the material (Table 1, samples 3, 4). Figure 12 shows Arrhenius plots of total conductivity in dry and wet air for two types of low-density ceramics. The first is hexagonal $La_{15}Mo_{8.5}O_{48}$ nanoceramic (relative density of 64.8%) produced by firing a mechanically activated $La_2O_3$ + $MoO_3$ mixture at 600 °C for 4 h. The second is the low-density tetragonal γ-$La_2MoO_6$ ceramic (relative density of 69.6%) produced by firing a mechanically activated $La_2O_3$ + $MoO_3$ mixture at 900 °C for 4 h. The impedance spectrum of the hexagonal $La_{15}Mo_{8.5}O_{48}$ nanoceramic is shown in Figure S4c. Like in the case of the hexagonal $La_{15}W_{8.5}O_{48}$ tungstate ceramics, there is no dispersion on the interface with the electrode, which suggests that electronic conduction prevails. In dry air, the activation energy for conduction (Table 2) in the $La_{15}Mo_{8.5}O_{48}$ molybdate nanoceramics (1.35 eV) corresponds to that in the $La_{15}W_{8.5}O_{48}$ tungstate nanoceramics (1.36 eV). Neither hexagonal $La_{15}Mo_{8.5}O_{48}$ nor tetragonal γ-$La_2MoO_6$ has proton conductivity.

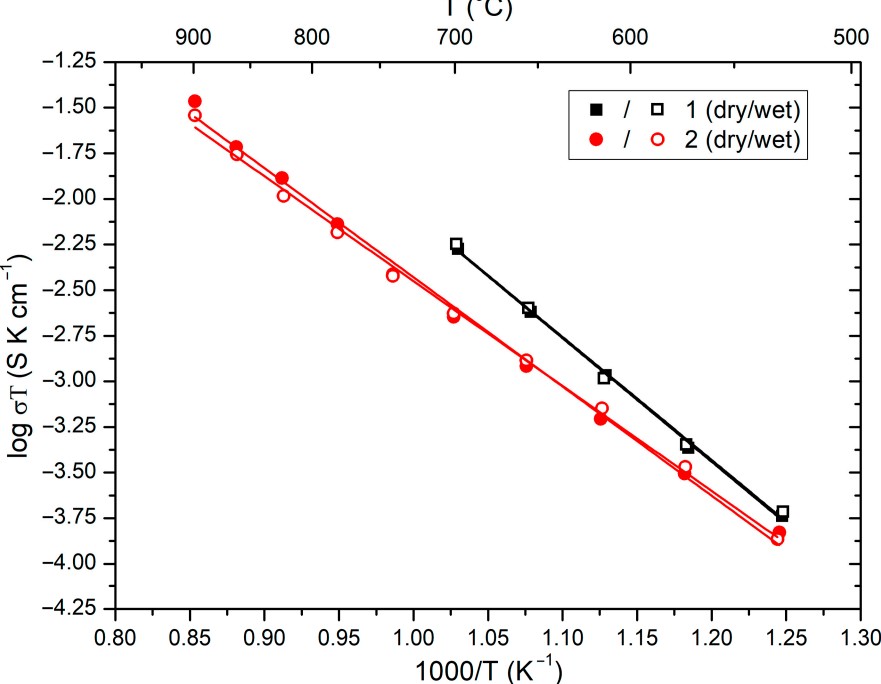

**Figure 12.** Arrhenius plots of total conductivity for LaMoO materials in dry (filled data points) and wet (open data points) air: (1) low-density hexagonal $La_{15}Mo_{8.5}O_{48}$ nanoceramic (relative density of 64.8%) produced by firing a mechanically activated $La_2O_3$ + $MoO_3$ mixture at 600 °C for 4 h; and (2) low-density tetragonal γ-$La_2MoO_6$ ceramic (relative density of 69.6%) produced by firing a mechanically activated $La_2O_3$ + $MoO_3$ mixture at 900 °C for 4 h.

**Table 2.** Apparent activation energy ($E_a$) of total conductivity temperature dependences of LaMoO materials in dry and wet air.

| No. | Composition, $T_{synthesis}$ | Atmosphere | $E_a$ ($\pm 0.01$), eV | |
|---|---|---|---|---|
| | | | 500–700 °C | 500–900 °C |
| 1 | $La_{15}Mo_{8.5}O_{48}$–600 °C | Dry air | 1.35 | – |
| | | Wet air | 1.34 | – |
| 2 | $\gamma$-$La_2MoO_6$–900 °C | Dry air | – | 1.14 |
| | | Wet air | – | 1.09 |

## 4. Conclusions

A hexagonal $La_{15}W_{8.5}O_{48}$ (phase II, polytype 5H ($P321$ no. 150)) nanophase has been synthesized for the first time via the firing of a mechanically activated $La_2O_3 + WO_3$ mixture at 800 and 900 °C for 6 h. The structure was identified by X-ray structure analysis with Rietveld refinement. At high temperatures (1000–1540 °C), a two-phase region has been found where the orthorhombic phase $\beta$-$La_2WO_6$ prevails and hexagonal $La_{18}W_{10}O_{57}$ is an impurity phase. DSC data for a mechanically activated $La_2O_3 + WO_3$ mixture demonstrate an exothermic peak due to the crystallization of the hexagonal phase $La_{15}W_{8.5}O_{48}$ from the X-ray amorphous state.

The mechanochemical synthesis of La molybdate is a considerably faster process than that of lanthanum tungstate. This is similar to what was previously reported for $La_{10}Mo_2O_{21}$ [46]. Short (4 h) isothermal heat treatments allowed us to obtain a hexagonal $La_{15}Mo_{8.5}O_{48}$ nanophase in the range of 500–700 °C. The DSC data for the mechanically activated $La_2O_3 + MoO_3$ starting mixture demonstrate two irreversible exothermic events: (1) crystallization of hexagonal $La_{15}Mo_{8.5}O_{48}$ (480–520 °C) and (2) irreversible first-order phase transition to the tetragonal phase $\gamma$-$La_2MoO_6$. The estimated crystallite size of hexagonal $La_{15}Mo_{8.5}O_{48}$ reaches ~50 nm (at 700 °C), and that of hexagonal $La_{15}W_{8.5}O_{48}$ is ~40 nm (at 900 °C). Both high-temperature phases ($\beta$-$La_2WO_6$ and $\gamma$-$La_2MoO_6$) were better crystallized.

Oxygen ion conductivity measurements for the dense, coarse-grained hexagonal $La_{18}W_{10}O_{57}$ (sp. gr. $P\overline{6}c2$, no. 190) ceramics have shown that oxygen ion conduction dominates only in the coarse-grained ceramics. By reducing the density and grain size of the material, electron conduction becomes dominant. In the low-density hexagonal $La_{15}Mo_{8.5}O_{48}$ nanoceramic (relative density of 64.8%) produced by firing a mechanically activated $La_2O_3 + MoO_3$ mixture at 600 °C for 4 h, electron conduction with an activation energy $E_a$ = 1.35 eV prevailed in dry air. This behavior is similar to what was observed in the high-temperature tetragonal phase $\gamma$-$La_2MoO_6$ ($E_a$ = 1.14 eV), as supported by the lack of electrode dispersion in its impedance spectrum.

Porous proton-conducting orthorhombic $\beta$-$La_2WO_6$ ceramics differing in relative density (67.1 and 81.4%) have been shown to have almost the same proton conductivity: ~$1 \times 10^{-6}$ S/cm at 600 °C. The proton conductivity appears to be independent of the density of the ceramics. No proton conduction has been detected in tetragonal $\gamma$-$La_2MoO_6$ ceramics.

**Supplementary Materials:** The following supporting information can be downloaded at: https://www.mdpi.com/article/10.3390/en16155637/s1, Table S1: Comparison of the refinement factors (Rexp, Rwp, Rp, $\chi^2$) for different models used in the Rietveld refinement of the XRD patterns obtained in the $La_2O_3 + WO_3$ mechanically activated oxide mixture with different thermal prehistory; Figure S1: Rietveld refinement of the XRD pattern of the m/a $La_2O_3 + WO_3$ mixture after heating to 760 °C. Crosses represent the observed pattern; the solid line through the symbols shows the calculated fit. Vertical bars mark the peak positions. The difference between the observed and calculated data is shown at the bottom; Table S2: Comparison of the refinement factors (Rexp, Rwp, Rp, $\chi^2$) for different models used in the Rietveld refinement of the hexagonal samples under consideration. The best models are highlighted in bold; Figure S2a: Data obtained in a DSC cell during heating

to 1200 °C and cooling for powder of nanocrystalline hexagonal $La_{15}W_{8.5}O_{48}$ (phase II polytype 5H) prepared as an intermediate phase by firing a mechanically activated $La_2O_3$ + $WO_3$ starting mixture at 800 °C for 6 h; Figure S2b: Data obtained in a DSC cell during heating to 1200 °C and cooling for powder of the coarse-grained hexagonal $La_{18}W_{10}O_{57}$ ceramic prepared by firing a mechanically activated $9La_2O_3$ + $10WO_3$ starting mixture at 1400 °C for 4 h; Figure S3: Microstructure of coarse-grained ceramics $La_{18}W_{10}O_{57}$, synthesized at 1400 °C for 4 h; Figure S4a: 700 °C impedance spectrum of the $La_{15}W_{8.5}O_{48}$ nanoceramic obtained as an intermediate phase by firing a mechanically activated $La_2O_3$ + $WO_3$ starting mixture at 800 °C for 6 h; Figure S4b: 700 °C impedance spectrum of the coarse-grained hexagonal $La_{18}W_{10}O_{57}$ ceramic produced by firing a mechanically activated $9La_2O_3$ + $10WO_3$ starting mixture at 1400 °C for 4 h; Figure S4c: 700 °C impedance spectrum of the $La_{15}W_{8.5}O_{48}$ nanoceramic obtained as an intermediate phase by firing a mechanically activated $La_2O_3$ + $MoO_3$ starting mixture at 800 °C for 6 h; Figure S4d: An equivalent electrical circuit used to fit the impedance spectra shown: (a)—in Figure S4a,c; (b)—in Figure S4b; Table S3: Density of the low-density ceramics under study.

**Author Contributions:** A.S., V.V. and E.B.: conceptualization, methodology, writing—original draft preparation, writing—review and editing; N.L., E.B., G.V. and O.K.: investigation, formal analysis; N.L., E.B., O.K. and D.S.: investigation, data curation, visualization; N.L., D.S. and I.K.: resources; N.L. and E.B.: formal analysis, writing—review and editing. All authors have read and agreed to the published version of the manuscript.

**Funding:** The support of this work from the Russian Foundation for Basic Research (Project 20-03-00399) is gratefully acknowledged. The work was supported partially by the subsidy from the Ministry of Education and Science allocated by the FRC CP RAS for the implementation of the state assignment (No. 122040500071-0, No. 122040500068-0) and in accordance with the state task for the FRC PCP and MC RAS, state registration No. AAAA-A19-119061890019-5. L.N.V. acknowledges the project of the HSE Scientific and Educational Group (No. 23-00-001).

**Data Availability Statement:** Data can be available upon request from the corresponding author and first author.

**Conflicts of Interest:** The authors declare no conflict of interest.

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
