# Peer review of "Synthesis of Hexagonal Nanophases in the La2O3–MO3 (M = Mo, W) Systems"

_energies, doi:10.3390/en16155637_

Round 1
Reviewer 1 Report
The authors reported the synthesis of hexagonal nanophases in the La2O3-MO3 (M = W, Mo) systems. The phases, microstructures, and conductivities of the products are thoroughly studied. This work is well designed and conducted. Something new deserved published in the journal of Energies. I have only two questions as listed below.
(1) In the introduction part, the authors said, two phases (6H and 5H) coexist in one crystal. I could not understand this sentence. Usually, there can be only one phase for one crystal.
(2) Can the authors explain the reason for the so-called dominated surface electronic conduction in the low-density hexagonal La15Mo8.5O48 nanoceramic?
Author Response
Reviewer 1
The authors reported the synthesis of hexagonal nanophases in the La2O3-MO3 (M = W, Mo) systems. The phases, microstructures, and conductivities of the products are thoroughly studied. This work is well designed and conducted. Something new deserved published in the journal of Energies. I have only two questions as listed below.
(1) In the introduction part, the authors said, two phases (6H and 5H) coexist in one crystal. I could not understand this sentence. Usually, there can be only one phase for one crystal.
Reply:
Thank you!
The phenomenon of polytypism in crystals has been known for a long time. To the best of our knowledge, the monograph by A. Verma, P. Krishna "Polymorphism and polytypism in crystals" was published by “Mir” in 1969.
Silicon carbide SiC, which for a long time was the only known polytype substance, has been studied in the most detail. ~ 25 different perfectly ordered structures with large periods (polytypes) have been discovered using X-ray methods. About 10 polytypes are known for ZnS. Polytypic substances include GdI2, KBr2, molybdenite MoS2, NbSe2.
In one single crystal, there are ordered structures with different periodicities (for example, the 5H or 6H polytypes in this work). Careful study of the complicated diffraction patterns revealed in each crystal sample two equally oriented hexagonal lattices with the common period a ~ 9.0 Å and different periods 6c and 5c, c ~ 5.4 Å . This indicated the co-existence of phases I and II (six- and five-layer polytypes) in one crystal, which differ in the character of the superstructure ordering [39].
[39] Novikova, N.E.; Sorokin, T.A.; Antipin, A.M.; Bolotina, N.B.; Alekseeva, O.A.; Sorokina, N.I.; Voronkova, V.I. Characteristic Features of Polytypism in Compounds with the La18W10O57-Type Structure. Acta Crystallographica Section C 2019, 75, 740–749, doi:10.1107/S2053229619006107.
Among the most recent works on the polytypism of crystals, the following is of particular interest: Alexander M. Antipin, Elena A. Volkova, Victor A. Rassulov , Nikolay N. Kuzmin, Elena Yu. Borovikova, Elena A. Latanova, Elizaveta V. Koporulina. A new double-cell polytype of samarium aluminum dimetaborate: Synthesis, crystal structure, and spectroscopic characterization. Materials Today Communications 31(2022) 103317.
https://doi.org/10.1016/j.mtcomm.2022.103317
New nonstoichiometric SmAl-dimetaborate crystals were synthesized using spontaneous nucleation from Sm2O3 – B2O3 – K2Mo3O10 flux melt. The room temperature and low-temperature single-crystal XRD investigations were carried out. Obtained solids crystallize in the non-centrosymmetric space group P 6¯2 m, with unit cells parameters a=b= 4.570(1) Å, c= 18.521(1) Å, V= 335.00(1) Å3 (for Sm2Al3.84B8O21.27, T = 295°K) and a=b= 4.563(1);Å, c= 18.505(1) Å, V= 333.91(1) Å3 (for Sm2Al3.96B8O21.30, T = 90°K), and z = 1. A comprehensive study of samarium double-cell polytype crystals was performed using scanning electron microscopy, luminescence spectroscopy and Raman measurements.
It is also important to note that the phenomenon of polytypism of crystals is rare in oxides.
(2) Can the authors explain the reason for the so-called dominated surface electronic conduction in the low-density hexagonal La15Mo8.5O48 nanoceramic?
Reply:
Thank you!
We suggest that in the more disordered La15W8.5O48 nanophase, oxygen transport pathways are hindered. This leads to a decrease in oxygen ion conductivity and a dominance of electron conductivity.

Reviewer 2 Report
This work presents the synthesis of nanophases La2O3-MO3 (M = Mo, W). The mixtures were characterized by DSC and XRD with Rietveld refinement, and SEM. The results presented in the work considerable importance and novelty in the subject field are published in this journal. In addition, discussion of the topic and associated literature is clear. However, the manuscript should be improved further before publishing
(1) Please add images of TEM nanophases.
(2) Please improve the quality of the figures in the revision.
Minor editing of English language required
Author Response
Reviewer 2
This work presents the synthesis of nanophases La2O3-MO3 (M = Mo, W). The mixtures were characterized by DSC and XRD with Rietveld refinement, and SEM. The results presented in the work considerable importance and novelty in the subject field are published in this journal. In addition, discussion of the topic and associated literature is clear. However, the manuscript should be improved further before publishing
(1) Please add images of TEM nanophases.
Reply:
Thank you!
We believe that in this case, the SEM study is sufficient to confirm the size of the nanoparticles in the ceramic material.
(2) Please improve the quality of the figures in the revision.
Reply:
We have improved the quality of the figures in the revision (Figs. 4b, 5, 6a, 6b, 7a, 7b, 8a, 9).

Reviewer 3 Report
In this work, the authors reported the synthesis of hexagonal nanophases in the La2O3-MoO3 and La2O3-WO3 systems. The synthesized mixtures were characterized using different analysis tools such as DSC, XRD, and SEM. The manuscript is well-written and the results are clearly presented. This paper needs only minor revisions to be accepted for publication. The authors should mention the full names of the abbreviations when it is first reported, i.e. DSC, XRD, and SEM that are written in the Abstract. Also, the English language should be carefully revised for typos and grammar mistakes.
The English language should be carefully revised for typos and grammar mistakes.
Author Response
Reviewer 3
In this work, the authors reported the synthesis of hexagonal nanophases in the La2O3-MoO3 and La2O3-WO3 systems. The synthesized mixtures were characterized using different analysis tools such as DSC, XRD, and SEM. The manuscript is well-written and the results are clearly presented. This paper needs only minor revisions to be accepted for publication. The authors should mention the full names of the abbreviations when it is first reported, i.e. DSC, XRD, and SEM that are written in the Abstract. Also, the English language should be carefully revised for typos and grammar mistakes.
Reply:
Thank you! We have changed the text.
DSC – Differential Scanning Calorimetry.
XRD – X-ray Diffraction.
SEM– Scanning Electron Microscopy.
Comments on the Quality of English Language
The English language should be carefully revised for typos and grammar mistakes.
Reply: Thank you! We have revised the text for typos and grammar mistakes.
